


# Mountain waves modulate the water vapor distribution in the UTLS

Romy Heller[1], Christiane Voigt[1,2], Stuart Beaton[3], Andreas Dörnbrack[1], Stefan Kaufmann[1], Hans Schlager[1], Johannes Wagner[1], Kate Young[3], Markus Rapp[1,4]

[1]Deutsches Zentrum für Luft- und Raumfahrt, Institut für Physik der Atmosphäre, Oberpfaffenhofen, Germany
[2]Johannes-Gutenberg-Universität Mainz, Institut für Physik der Atmosphäre, Mainz, Germany
[3]National Center for Atmospheric Research, Boulder, Colorado, USA
[4]Ludwig-Maximillians-Universität München, Meteorologisches Institut München, Munich, Germany

*Correspondence to*: Christiane Voigt (Christiane.Voigt@dlr.de)

**Abstract.** The water vapor distribution in the upper troposphere/lower stratosphere region (UTLS) has a strong impact on the atmospheric radiation budget. Transport and mixing processes on different scales mainly determine the water vapor concentration in the UTLS. Here, we investigate the effect of mountain waves on the vertical transport and mixing of water vapor. For this purpose we analyse measurements of water vapor and meteorological parameters recorded by the DLR Falcon and NSF/NCAR GV research aircraft taken during the Deep Propagating Gravity Wave Experiment (DEEPWAVE) in New Zealand. By combining different methods, we develop a new approach to quantify location, direction and irreversibility of the water vapor transport during a strong mountain wave event on 4 July 2014. A large positive vertical water vapor flux is detected above the Southern Alps extending from the troposphere to the stratosphere in the altitude range between 7.7 and 13.0 km. Wavelet analysis for the 8.9 km altitude level shows that the enhanced upward water vapor transport above the mountains is caused by mountain waves with horizontal wavelengths between 22 and 60 km. A downward transport of water vapor with 22 km wavelength is observed in the lee-side of the mountain ridge. While it is a priori not clear whether the observed fluxes are irreversible, low Richardson numbers derived from dropsonde data indicate enhanced turbulence in the tropopause region related to the mountain wave event. Together with the analysis of the water vapor to ozone correlation we find indications for vertical transport followed by irreversible mixing of water vapor.

For our case study, we further estimate greater than 1 W m$^{-2}$ radiative forcing by the increased water vapor concentrations in the UTLS above the Southern Alps of New Zealand resulting from mountain waves relative to unperturbed conditions. Hence, mountain waves have a great potential to affect the water vapor distribution in the UTLS. Our regional study may motivate further investigations of the global effects of mountain waves on the UTLS water vapor distributions and its radiative effects.

## 1 Introduction

Water vapor is a major greenhouse gas in the UTLS (Sherwood et al., 2010; Solomon et al., 2010). Thus, changes in the water vapor distribution in the UTLS cause a radiative forcing and may affect surface temperatures (Solomon et al., 2010; Riese et


al., 2012). Therefore, understanding of sources and sinks as well as transport and mixing of water vapor (Holton et al., 1995; Gettelman et al., 2011) is fundamental to quantify its impact on the atmospheric radiation budget.

There are a few studies that refer to trace gas transport induced by gravity waves (e.g. Danielsen et al., 1991; Langford et al., 1996; Schilling et al., 1999; Moustaoui et al., 2010). Gravity waves are known to play an important role for circulation, structure and variability of the atmosphere (Fritts & Alexander, 2003). They distribute energy and momentum horizontally and vertically in the atmosphere (e.g. Smith et al., 2008; Geller et al., 2013; Wright et al., 2016). The vertical displacement of an air parcel by gravity waves creates fluctuations in trace gas concentrations at constant altitude if the trace gas distribution has a vertical gradient (Smith et al., 2008). For adiabatic processes tracer mixing ratios as well as the potential temperature are thereby conserved. With respect to an analysis of an adiabatic process, water vapor may serve as an excellent tracer for gravity waves in the troposphere to the lower stratosphere region, while for example ozone is a good tracer for the stratosphere. Previous studies investigated the effects of gravity waves on the ozone or carbon monoxide distribution (e.g. Langford et al., 1996; Teitelbaum et al., 1996; Schilling et al., 1999; Moustaoui et al., 2010), while the effects on water vapor are less discussed due to the complex interaction of sources and sinks of water vapor in the UTLS region, for example the possibility of condensation (Moustaoui et al., 1999; Pavelin et al., 2002). Schilling et al. (1999) measured strong fluctuations in CO mixing ratios at a constant flight level (11.9 km) caused by mountain waves and calculated the vertical trace gas flux at this altitude. They derived an upward transport of CO that resulted in enhanced CO mixing ratios at a higher altitude (12.5 km). They speculated that dynamic instabilities were induced by wave breaking and that convective overturning finally led to an irreversible vertical CO transport.

The method to calculate the vertical trace gas flux (Shapiro, 1980; Schilling et al., 1999) is similar to the calculations of energy and momentum fluxes (e.g. by Smith et al., 2008) and indicates the vertical transport direction of the trace gas. If we assume a negative gradient for the trace gas, a positive flux generally will indicate an upward transport of high mixing ratios into a region with low mixing ratios. However, it may also display a downward transport from a region of low mixing ratios to a region with higher mixing ratios (e.g. by existence of an inversion layer). The transport of trace gas species may be reversible or irreversible, depending on mixing processes on different scales. Irreversible mixing is promoted by turbulence induced, for example, by nonlinear wave interaction, wave breaking, or dissipation (Lamarque et al., 1996; Whiteway et al., 2003; Koch et al., 2005; Lane & Sharman, 2006). Danielsen et al. (1991) showed that waves with large horizontal wavelengths (~36 – 270 km) and enhanced vertical amplitudes are significant carriers of energy, momentum and trace species. Small-scale waves (horizontal wavelength smaller than 30 km) may cause mixing and thus enable the irreversibility of the transport induced by large-scale waves. In a later study, Moustaoui et al. (2010) showed that small-scale waves can also be effective in transport based on reversible dynamic processes.

One method to investigate mixing of trace gases in the UTLS region is to consider the correlation between a tropospheric and a stratospheric tracer (e.g Fischer et al., 2000; Hoor et al., 2002; 2004; Pan et al., 2007). In an idealized non-mixed atmosphere, a tropospheric tracer (e.g. $H_2O$) and a stratospheric tracer (e.g. $O_3$) are not correlated and show a "L-shape" in a 2D tracer-tracer-plot. Mixing processes across the tropopause (for example by troposphere-stratosphere-transport related to tropopause folds or





convection) can lead to linear relations (mixing lines) between the tracers. This feature is observed only for irreversible mixing. The strength of the mixing and thus the slope of the mixing line is a function of the tracer distributions in the initial air mass and the elapsed time since the mixing took place (Hoor et al., 2002). The tracer-tracer correlation are based on a dynamic approach but may be affected by microphysics in the case of water vapor. The consequences of condensation in the tropopause region are not completely displayed in such a correlation plot.

The objective of this paper is to investigate transport of water vapor during a strong mountain wave event using a new combination of different techniques common in gravity wave and atmospheric transport analysis. While previous studies focused on single altitudes, we use measurements in the altitude range between 7.7 km and 13.0 km to cover the upper troposphere and lower stratosphere including the tropopause region. In contrast to previous studies, which mainly applied simulations (Schilling et al., 1999; Moustaoui et al., 2010), here we investigate the irreversibility of the water vapor transport by using in-situ information from tracer-tracer correlations and vertical dropsonde profiles. Furthermore, we are interested in a possible impact of the water vapor distribution in the UTLS on the radiation budget based on radiative transfer calculations by Riese et al. (2012).

To this end, we analysed measurements from three research flights of the DLR Falcon 20E and the NSF/NCAR Gulfstream V (GV) research aircraft during the DEEPWAVE (Deep Propagating Gravity Wave Experiment) campaign in June/July 2014 above New Zealand (Fritts et al., 2016). The campaign focussed on a better understanding of the life cycle of gravity waves from excitation and propagation to dissipation at high altitudes. For the first time, DEEPWAVE combined ground-based and airborne measurements as well as satellite observations over New Zealand and the Southern Pacific – a "hotspot" region for gravity waves during the southern hemispheric winter. Here, we show results from measurements on 4 July 2014 taken during a strong mountain wave event over the Southern Alps.

First, we describe the in-situ measurements on the DLR Falcon and NSF/NCAR GV and the methods to investigate the water vapor transport induced by the mountain waves. Next, we present results for the vertical water vapor flux on a specific flight leg in the upper troposphere. Corresponding wavelet spectra reveal the location and scales of the vertical fluxes. This is followed by a general discussion of the fluxes over a wide altitude range. We then use dropsonde data to identify turbulence layers and investigate tracer-tracer correlations to quantify mixing along the flight tracks over the mountains. Finally, we discuss the effects of the mountain waves on the water vapor distribution in the UTLS and on atmospheric radiative transfer.

## 2 Instrumentation

During the DEEPWAVE campaign, the Falcon and the GV were equipped with a set of in-situ instruments to determine the trace gas composition and meteorological parameters. Here, we describe the instruments with relevance to this work.

### 2.1 Frost point hygrometer on the Falcon

The gas phase water vapor mixing ratio was determined with the cryogenic frost point hygrometer CR-2 (Buck Research Instruments, LLC) (Voigt et al., 2010; 2011). The instrument measures the temperature of a mirror covered with a thin frost





layer that is kept in thermal equilibrium with the ambient water vapor in a closed cell. An optical detector determines the thickness of the frost layer by measuring its reflectivity. The mirror is temperature-regulated so that the condensate layer thickness remains constant. In that state the mirror temperature equals the ambient frost/dew point temperature. Then, the water vapor mixing ratio can be calculated using the inverse Clausius-Clapeyron equation. The instrument covers a wide

measurement range between 1 and 20,000 ppmv suitable for tropospheric and stratospheric conditions. The sampling time of the CR-2 hygrometer is 0.3 Hz. The data are quality checked by calibrations before and after the campaign against a reference MBW 373LX dew point mirror. During previous campaigns (Voigt et al., 2010; Voigt et al., 2014), the instrument agreed well (within ±10 %) with high accuracy water vapor data measured with the airborne mass spectrometer AIMS-$H_2O$ (Kaufmann et al., 2014; 2016). For this campaign, an additional correction for low water vapor mixing ratios has been derived from

simultaneous water vapor measurements on the GV research aircraft. The uncertainty of the water vapor mixing ratios is determined by systematic errors in the temperature measurements of the mirror and by the calibration accuracy. The uncertainty is 9 to 12 % for water vapor mixing ratios between 10 and 500 ppmv (Table 1). In the troposphere, the response time of the CR-2 to sudden changes in the mixing ratio is on the order of one to a few seconds. In the stratosphere, the absolute change in water vapor mixing ratios is smaller but the response time can be longer because the time to equilibrate the mirror

temperature is longer for low mixing ratios. Therefore, the amplitudes in the CR-2 water vapor measurements in the stratosphere (<10 ppmv) may be damped and thus these data are not used quantitatively in this study.

## 2.2 Ozone measurements on the Falcon

Ozone was measured by an ultraviolet (UV) photometric gas analyser TE49 (Thermo Environmental Instruments, Inc.) (Schumann et al., 2011; Huntrieser et al., 2016). The absorbance at the wavelength of 254 nm is directly related to the ozone

concentration by the Beer-Lambert law. The sampled air is split into two gas streams which flow to separate optical measurement cells. The gas in one cell serves as reference after ozone is removed by a scrubber. The two cells allow for a simultaneous measurement of both gas streams. The flow to the cells is alternated every 4 s using a selenoid valve. The response time is 15 s with a lag time of 10 s. The precision and accuracy are 1 ppbv and ±5 %, respectively.

## 2.3 Meteorological parameters on the Falcon

The DLR Falcon aircraft carried a basic meteorological instrumentation suite (Krautstrunk & Giez, 2012). Sensors for pressure and wind are located at the noseboom to realise undisturbed measurements of the ambient air. Total Air Temperature (TAT) was measured with an open wire PT100 sensor at the bottom fuselage in the front. To obtain the static air temperature the TAT has to be corrected by the Mach number of the aircraft. The measurement uncertainty is ±0.5 K. The wind speed is derived by a differential pressure sensor in combination with GPS data for aircraft position and orientation. For the horizontal wind

components the measurement uncertainties are ±0.7 m s$^{-1}$ (along wind component) and ±0.9 m s$^{-1}$ (cross wind component) and for vertical wind ±0.3 m s$^{-1}$. Flight altitudes are determined by a barometer as part of the inertial reference system as well as by GPS tracking. All measurements are stored with a time resolution of 1 Hz.



## 2.4 The laser hygrometer and dropsonde measurements on the GV

The NSF/NCAR GV aircraft was also equipped with instrumentation to obtain the meteorological parameters at 1 Hz time rate (Fritts et al., 2016; Smith et al., 2016). Water vapor measurements were made by an open path Vertical Cavity Surface Emitting Laser hygrometer (VCSEL, Southwest Sciences, Inc.) (Zondlo et al., 2010). The instrument was installed on the

bottom fuselage of the aircraft. Two water vapor absorption lines are used to cover a wide measurement range of high mixing ratios (1853.3 nm) and moderate to low mixing ratios (1854.0 nm). The uncertainty is ±5 % at a sampling rate of 25 Hz.

On 4 July, 16 dropsondes were launched by a fully automated Airborne Vertical Atmospheric Profiling System (AVAPS) (Young et al., 2014). The dropsondes (Vaisala, Inc.) contain sensors to measure atmospheric temperature, pressure and humidity and a GPS receiver to derive winds. The data were stored at 2 Hz which provides a vertical resolution of less than

10 m in the atmosphere. The uncertainties for the temperature are ±0.2 K and for the horizontal winds ±0.5 m s$^{-1}$.

## 3 Methods

We present a novel combination of methods to analyse trace gas transport induced by mountain waves. First, we calculate the vertical water vapor flux $\overline{w'q'}$ in the measurement region as a general transport parameter correlated to the vertical wind motion. Further, the wavelet analysis of $\overline{w'q'}$ reveals the location, the wavelength and the direction of the vertical trace gas

transport. Generally, the method can be applied to any conservative tracer with a gradient in the troposphere and/or the stratosphere. Thus, we apply it in this study to a flight in cloud-free conditions. Finally, we investigate the reversibility of the transport using dropsonde data and tracer-tracer correlations.

### 3.1. Choice of case study and data preparation

On 4 July 2014, two flights were performed with the DLR Falcon (referred to as flight numbers FF04 and FF05) and one flight

with the NSF/NCAR GV (flight number RF16) (Table 2). On that day, classified as intensive observation period (IOP) number 10 of the DEEPWAVE campaign, a strong mountain wave event with the highest vertical wave-induced energy fluxes during the whole campaign occurred (Fritts et al., 2016; Smith et al., 2016).

During IOP 10 a south-westerly and west-south-westerly flow over the South Island of New Zealand reached more than 40 m s$^{-1}$ (Figure 1a). The flight pattern of the DLR Falcon (Figure 1) was chosen to be nearly parallel to the main wind

direction over the Southern Alps with Mt. Aspiring as highest summit. The flight legs above the mountains were flown four times during each Falcon flight to cover different altitudes and to determine the mountain wave situation below and above the tropopause. For the analysis, we define cross sections covering the whole mountain range at each altitude with neither an altitude change nor a turn of the aircraft. Mt. Aspiring was defined as the reference point for each flight leg and the distance to this reference point is used as x-axis scaling for the analysis.

In this study, we use all data in a 1 Hz time resolution. Therefore, we interpolated the lower frequency measurements to a one second time grid as consistent data input for a subsequent wavelet analysis. A sensitivity analysis using the coarser time





resolution of the CR-2 (0.3 Hz) as base for the evaluation did not change the results. Since we are not looking at this point into the turbulent part of the wavelength spectrum, the 1 s-interpolation is sufficient.

## 3.2 Method to calculate the vertical water vapor flux

The calculation of a vertical trace gas flux was first described by Shapiro (1980). The basic assumption of this method is that
we consider a conservative and passive tracer $q$ without sources and sinks such that:

$$\frac{dq}{dt} = \frac{\partial q}{\partial t} + \vec{v} \cdot \overrightarrow{\mathrm{div(q)}} = 0, \tag{1}$$

where $\vec{v}$ is the vector field of the horizontal and vertical wind components $u$, $v$ and $w$ and $\overrightarrow{\mathrm{div(q)}}$ is the divergence of the passive tracer. Since there were no clouds over the mountain transect for the analysed altitudes, water vapor is a conservative tracer in our case and its distribution is not influenced by condensation or sublimation.

The quantity $q$ as well as the wind components may be expressed in terms of a spatial or temporal mean $\bar{q}$ and perturbations $q'$:

$$q(x) = \bar{q} + q'(x). \tag{2}$$

Under the assumption that we can neglect the mean horizontal and vertical advection of $\bar{q}$ in the measurement region we consider the local temporal change of $\bar{q}$ as:

$$\frac{\partial \bar{q}}{\partial t} = -\frac{\partial}{\partial x}(\overline{u'q'}) - \frac{\partial}{\partial y}(\overline{v'q'}) - \frac{\partial}{\partial z}(\overline{w'q'}), \tag{3}$$

where $u'$ and $v'$ are the horizontal wind perturbations and $w'$ is the vertical wind perturbation. The overbars mark the mean trace gas flux over spatial or temporal intervals. Furthermore, we assume that the horizontal flux divergences are negligible compared to the vertical flux divergence and hence eq. (3) reduces to:

$$\frac{\partial \bar{q}}{\partial t} = -\frac{\partial}{\partial z}(\overline{w'q'}). \tag{4}$$

The local vertical trace gas flux is then determined by

$$w'q'(x) = q'(x) \cdot w'(x), \tag{5}$$

where the perturbations depends on the filter function used to receive the spatial mean:

$$\bar{q} = \frac{1}{x_2 - x_1} \cdot \int_{x_1}^{x_2} q(x)dx. \tag{6}$$

We derive the mean vertical trace gas flux $\overline{w'q'}$ from integrating over selected spatial or temporal intervals along the mountain
cross section (see section 4.4). For an ideal linear wave, the mean vertical flux would be zero. If we observe a negative or positive mean flux, a trace gas transport will exist but we need further analysis on the irreversibility of the transport process.

The filter function and its characteristics have an important influence on the results. We have to decide which scales of horizontal wavelength to include and which parts of the spectrum to neglect. In this study, the length of the flight legs limits the maximum resolvable horizontal wavelength to less than 150 km. In addition, a change in the wind direction in front of the
mountains partially influences the water vapor distribution at wavelengths larger than 80 km. Therefore, a suitable filter choice in our case is a bandpass filter with a lower limit of 300 m and an upper limit of 80 km. It must be kept in mind that the





bandpass filter as spatial filter method may damp wavelengths due to edge effects (Ehard et al., 2015). We apply the filter to the water vapor measurements as well as to the vertical wind measurements.

### 3.3 Wavelet analysis method

Wavelet analysis is widely used in gravity wave analysis to identify the location and wavelength scale of waves (e.g. Woods & Smith, 2010; Placke et al., 2013; Zhang et al., 2015). By combining power spectra and cospectra of the variables of interest, flux carrying waves can be characterized. We calculated normalized power spectra of the vertical wind and the water vapor perturbation using the Morlet wavelet as defined in Torrence & Compo (1998) and an equalized distance of 200 m between each data point. For the calculation we create standard normal distributed perturbed variables $q'$ and $w'$.

The cospectrum $W_n^{XY}(s)$ of the vertical water vapor flux $w'H_2O'$ ($q' \equiv H_2O'$) combines the real parts of the wavelet spectra of both variables:

$$W_n^{XY}(s) = \Re\{W_n^X(s)W_n^{Y^*}(s)\}, \tag{7}$$

where $X$ and $Y$ represent the variables $w$ and $H_2O$, $n$ classifies the localised position index, $s$ is the wavelet scale and $^*$ is the complex conjugate. This results in the in-phase contributions to a product from different wavelengths. The significance is determined with the method from Portele (2016) as follows:

$$\frac{\left|W_n^X(s)W_n^{Y^*}(s)\right|}{\sqrt{|P_k^X P_k^Y|}} = \frac{\chi_\nu^2(p)}{\nu}, \tag{8}$$

where $P_k$ represents the normalized Markov red noise spectrum with the frequency index $k = 0 \ldots N-1$ with $N$ as the number of points in the data series, $\chi_\nu^2$ is the chi-square distribution for $\nu$ degrees of freedom and $p$ is the significance. For this case we use in eq. (8):

$$P_k = \frac{1-\alpha^2}{1+\alpha^2 - 1\alpha\cos(2\pi k/N)}, \tag{9}$$

a combined autocorrelation factor $\alpha$ with a lag of one and a lag of ten ($\alpha = lag1 + \sqrt{lag10}/2$) that gives defined significant areas with uniform weighting of low and high fluxes over the wavelength scale.

### 4 Results

First, we show the results of the flux calculations and wavelet analysis for one selected flight altitude. Then, we discuss the water vapor measurements on different flight altitudes to characterize the vertical flux from the upper troposphere to the lower stratosphere. Finally, we use dropsonde data to identify regions with enhanced turbulence and with a vertical gradient of the potential temperature close to zero. Additionally, we investigate mixing processes in the measurement region using tracer-tracer correlations, in this case of water vapor and ozone.



## 4.1 Synoptic situation on 4 July 2014

Mesoscale simulations with the Weather Research and Forecasting (WRF) model, version 3.7 (Skamarock et al., 2008) were performed to give an overview of the synoptic situation. Two nested domains with horizontal resolutions of 6 km and 2 km and 138 vertical levels with a model top at 2 hPa were used. The model is initialized with operational analyses of the ECMWF model at 18 UTC on 3 July 2014 and run for 36 hours. A detailed overview of the same model set-up including the parameterizations used can be found in Ehard et al. (2016) for a gravity wave event over northern Scandinavia.

For 4 July 2014 the orographic forcing over the Southern Alps was induced by a south-westerly wind of ~20 m s$^{-1}$ at 850 hPa at the west coast of the South Island of New Zealand (not shown). Up to the tropospheric jet level around 8.9 km (at 300 hPa), horizontal wind speeds in the upstream region accelerate up to 50 m s$^{-1}$ (Figure 1a). Over the mountains the horizontal wind velocities decreased to 30 to 40 m s$^{-1}$ (Figure 1b) and changed from a westerly direction west of the South Island to south-westerly. A part of the core region of the tropospheric jet was located west of the South Island (Figure 1b). The strong low-level flow forced mountain waves as clearly indicated by the vertical wind speed values over the island at 8.9 km altitude (Figure 1c).

The intensity of the mountain wave forcing over New Zealand on 4 July 2014 changed within several hours. The forcing at the west edge of the mountains was strongest at 06 UTC and weakened till 18 UTC. Also, a low pressure system south of New Zealand moved quickly eastward and led to a thermal tropopause (WMO, 1957) descending from 11.1 km to 9.5 km during the observation period. A detailed overview of the synoptic situation for 4 July 2014 is given by Bramberger (2015).

## 4.2 Vertical water vapor flux at 8.9 km

An overview of the first Falcon flight FF04 on 4 July 2014 is shown in Figure 2. We identify strong fluctuations in water vapor, potential temperature and horizontal and vertical wind components at different altitudes during the flight. In particular, the vertical wind component varied ±5 m s$^{-1}$ over the mountains (bottom panel). For water vapor we detect the strongest perturbations in the same region over the mountains during the first and second flight leg (7.7 and 8.9 km) with amplitudes of up to 100 ppmv. The amplitudes decrease with altitude due to the general decline of the H$_2$O concentrations in the UTLS. Ozone shows strong variations over the mountains in the stratosphere and less variability in the troposphere, opposite to the H$_2$O signal. Also, the potential temperature as well as the horizontal wind components displays fluctuations above the mountains. The location and extent of the fluctuations imply mountain waves as source as suggested by studies from Smith et al. (2016) and Bramberger (2015).

For this work we chose the second flight leg at 8.9 km as an example to analyse the water vapor transport (Figure 3). In the upper troposphere the water vapor measurements with the CR-2 hygrometer are very sensitive to sudden changes in the mixing ratio as caused by mountain waves. The flight leg is located in the upper troposphere with a distance of approximately 2 km to the thermal tropopause at 10.9 km. The wave signature in water vapor is very distinctive with high amplitudes of 20 ppmv above the Mt. Aspiring transect. The potential temperature shows a similar wave pattern as water vapor, but anti-correlated





(selected instances indicated by vertical blue dashed lines in Figure 3) and following the vertical wind fluctuations. Additionally, there is a slow decrease in the water vapor mixing ratio from -80 km distance to the summit at $x = 0$, along with an increase in the potential temperature of 3 K and a change in the wind direction. The upstream region of the transect is located in the vicinity of the tropospheric jet stream which may influence the upstream water vapor distribution by horizontal larger-scale processes.

Results from the flux calculations for this flight leg are displayed in Figure 4. We applied the bandpass filter with an upper limit of 80 km wavelength to the water vapor and the vertical wind data and show the received perturbations $w'$ and $H_2O'$ in panel (a). The two variables are 90° phase shifted with respect to each other which can be also observed by the diagonal blue dashed lines in the bottom panel of Figure 3. This phase shift is caused by a direct response of water vapor to the vertical wind motion. We assume an atmosphere finely layered with conserved quantities. These layers are disturbed by propagating gravity waves and an aircraft flying at a constant level penetrates the layers repeatedly as depicted in Figure 10 of Smith et al. (2008). At a constant altitude therefore the trace gas concentration and potential temperature follow the vertical wind variations with a phase shift of 90°.

A strong wave signature is detected in the local vertical water vapor flux $w'H_2O'$ above the mountains (Figure 4b). The vertical flux is very small in the upstream region and the amplitude increases over the mountains from west to east. Figure 4c shows the integrated vertical water vapor flux. It is generally positive above the mountains between +30 and +60 km and from +90 to +180 km distance to the Mt. Aspiring summit with a maximum of 39,000 m² s⁻¹ ppmv and 76,000 m² s⁻¹ ppmv, respectively. Further east we find a negative trend (-176 m ppmv) induced by little water vapor perturbations but enhanced vertical wind fluctuations.

Since water vapor has a negative gradient in the troposphere, a positive flux mainly indicates upward transport of high mixing ratios to a level of lower mixing ratio. A negative flux points to a downward transport. Thus, we find a strong indication of an integrated upward water vapor flux above the Southern Alps and a downward flux above the eastern part of the mountains for this flight leg.

For the flux calculations we defined water vapor as conservative tracer due to the absence of supersaturation at the analysed flight altitudes. However, at the first flight leg of FF04 at 7.7 km we measured ice particles between +150 km and +200 km distance that indicate the existence of lee wave cirrus. These gravity wave induced clouds were also visible in the infra-red images of the MTSAT-2 satellite at 03 UTC and dissipated until 06 UTC (Bramberger, 2015). This may affect the water vapor distribution at the next flight level at 8.9 km by lowering the amplitude of the fluctuation. In Figure 3 we observe a strong peak in the vertical wind at +170 km distance to the summit in contrast to a small water vapor fluctuation which may be influenced by the drying of the level below. The calculated flux in this region is then also reduced. This effect does not influence the general transport direction at this flight altitude and is not relevant for the higher flight altitudes or the second Falcon flight since these lee wave clouds dissipated during the first flight.



### 4.3 Wavelength spectrum of the vertical water vapor flux

Wavelet analysis is used to quantify location, scale and direction of the vertical water vapor flux. Figure 5 shows the amplitudes of perturbations in vertical wind (a) and water vapor (b) for the second flight leg of FF04 for horizontal wavelengths between 300 m and 400 km. The power spectra represent the variance wavelets for $w'$ and $H_2O'$ while the

cospectrum shows the covariance wavelet for $w'H_2O'$. Highest activity in both variables occurs for wavelengths between 10 km and 80 km, where the upper limit results from the bandpass filter. Moreover, the peaks are located above the middle and eastern part of the mountains. We find similar patterns in $w'$ and $H_2O'$ but of different intensity. Water vapor has the strongest peak at +75 km distance and at 22 km horizontal wavelength, whereas the intensities of the vertical wind perturbation are strongest further east at +180 km from the summit with a broader wavelength range between 15 and 30 km. The power of the

water vapor fluctuation in this region may be reduced due to condensation at the flight altitude below as mentioned before. Since the flight legs are short in the downstream region, data for $x > +200$ km lie in the cone of influence (COI) area and thus require careful interpretation due to edge effects of the analysis (Torrence & Compo, 1998). Additionally, we find a layer of enhanced magnitude in the power of the water vapor perturbations at a wavelength of about 60 km located at -80 to +100 km distance. This may be caused by longer waves that are not part of this analysis and that are influenced by horizontal advection

due to the tropospheric jet stream. There are some significant areas in the upstream region as well as over the eastern part of the mountains in both power spectra for wavelengths <10 km with amplitudes <0.1 $m^2\,s^{-2}$ and <0.1 $ppmv^2$, respectively. This indicates additional small scale fluctuations in the parameters that may not be relevant for transport of water vapor but for mixing processes. These small scale fluctuations are especially observed for the vertical wind over the middle and eastern mountain region at the higher altitudes (not shown) where we find indications for turbulence in the dropsonde data (see section

20   5).

In the right panels of Figure 5, we show the global wavelet spectrum (GWS) where the power is averaged over all local wavelet spectra. This highlights the dominant wavelengths along the flight path. Most power is carried in wavelengths smaller than 30 km for both variables. A second mode with less power is found between 40 and 80 km horizontal wavelength.

Figure 5c shows the corresponding cospectrum of $w'H_2O'$. As in the individual power spectra we identify dominant horizontal

wavelengths between 10 and 80 km. The location of upward or downward transport is represented in the colour-coding with red areas indicating an upward $H_2O$ flux and blue the opposite. The significant parts from the individual power spectra contribute to the local flux. Horizontal wavelengths between 22 and 60 km dominantly contribute to an upward water vapor transport above the mountain region. The downward water vapor flux above the eastern mountain part is mainly carried by wavelengths between 20 and 22 km. The vertical wind perturbation dominantly influences this transport direction. Quadrant

analysis of $w'$ and $H_2O'$ (not shown) reveal that the positive flux $w'H_2O'$ is dominated by the upward transport of high humidity in regions with low humidity for wavelengths larger than 22 km. Less pronounced is the downward transport of low humidity that also cause a positive flux. The negative flux for horizontal wavelengths smaller than 22 km is a result of the





upward transport of low humidity and the downward transport of high humidity in equal parts which caused a reduced water vapor mixing ratio in this region.

The results show an overall upward transport of $H_2O$ at this flight altitude. Further, a superposition of wave packets with different characteristics is detected in the mountain wave region. The rugged terrain of the Southern Alps with many crests and

valleys may initiate these different contributions to the full spectrum. In the statistical analysis of all GV flight level data during DEEPWAVE, Smith et al. (2016) also observed small and longer scale waves with different characteristics. In their study flux-carrying waves are larger than 20 km horizontal wavelength. Small scale waves with wavelengths around 20 km and less are mainly dominating in the vertical wind motion and do not carry any energy or momentum flux upward (Smith & Kruse, 2017).

**4.4 Vertical profile of the water vapor flux from the troposphere to the stratosphere**

We combine GV and Falcon data on 4 July 2014 to derive a profile of the vertical water vapor flux in the UTLS region. The Falcon flights FF04 and FF05 covered a temporal evolution of the mountain wave activity that increased from the first to the second flight (Bramberger, 2015). The GV operated simultaneously to the second Falcon flight FF05 (Table 2). Both aircraft flew on the same flight track but at different altitudes to measure the vertical propagation of the mountain waves. In Figure 6a,

we show the Falcon flight legs 1 to 3 (FF05) between 7.7 and 10.8 km and two GV flight legs at 12.0 and 13.0 km that took place at the same time as leg 3 and leg 4 of FF05. The fourth leg of FF05 is not shown since amplitudes in the water vapor fluctuations cannot be fully resolved by the CR-2 in the stratosphere. During all Falcon and GV transects, we find significant water vapor fluxes over the mountain region (Figure 6a). The thermal tropopause was located at about 10.5 km, thus the observed water vapor flux extends above the tropopause. The wave pattern remains nearly stationary through all altitudes with,

for example, a strong wave package at about +110 km distance from the reference point. Upstream and downstream regions exhibit very low or no vertical fluxes.

To derive a vertical profile of the vertical water vapor flux in the mountain wave region, we define a range between the highest summit ($x = 0$ km) and the east end of the Southern Alps ($x = 202$ km). For this region, the integrated vertical water vapor flux is normalized by the length. The results at each altitude are plotted in Figure 6b. To distinguish the transport characteristics of

different horizontal wavelengths, we show the profile for wavelengths between 300 m and 80 km and between 22 km and 80 km, respectively. Under the assumption of quasi-stationary mountain waves, we neglect the time shift (~3 h) between the single flight legs.

In general, a negative (positive) flux divergence $\frac{\partial(\overline{w'H_2O'})}{\partial z}$ humidifies (dries) the layer above due to the negative water vapor gradient in the atmosphere. By the absence of vertical or horizontal transport and the existence of a well-mixed atmosphere, we

are expecting no flux divergence. In the mountain region we see positive flux divergences in the troposphere (7.7 – 8.9 km) and lower stratosphere (10.8 – 12.0 km) for horizontal wavelengths between 300 m and 80 km (Figure 6b) which may indicate a general downward transport (Table 3). The strong negative vertical flux at the lowest altitude may be influenced by transport





and mixing processes that lie below this level and that are not covered by the in-situ measurements. This may be convective processes in front or over the mountains. The positive flux divergence in the layer from 10.8 to 12 km implies a drying of the atmosphere by a downward transport. In the layer below, from 8.9 to 10.8 km, a strong upward transport from the upper troposphere through the tropopause occurs which is indicated by the negative flux divergence. This process may lead to the observed enhanced water vapor mixing ratios at around 10.8 km and below (see section 6). Since we only have measurements on a few defined altitudes an exact localisation of maxima and of sign changes of the transport direction is not possible.

The picture changes when excluding the small wavelengths below 22 km (dashed line in Figure 6b). We then find a negative flux divergence over the broad altitude range from upper troposphere to lower stratosphere (8.9 – 13 km). This indicates a dominating upward transport of water vapor by the larger wavelengths (see Figure 5c). When comparing both profiles, the difference between them in the layer between 8.9 and 12.0 km suggests that the positive flux divergence (downward transport) between 10.8 and 12.0 km is mainly induced by small horizontal wavelengths (Table 3). These smaller wavelengths indicate instabilities in the atmosphere and thus the upward mountain wave propagation may be influenced by local turbulence (see section 5) or by downward propagating gravity waves that are excited aloft (Bramberger, 2015).

## 5 Turbulence in the UTLS region

Gravity waves may cause or enhance turbulence by instabilities, wave breaking, and dissipation (e.g. Pavelin et al., 2002; Fritts & Alexander, 2003; Whiteway et al., 2003). Here, we use dropsonde launches from the GV to investigate turbulence potentially induced by the mountain waves which may cause mixing of trace species in the measurement region. Therefore, we calculate potential temperature and Richardson numbers (Ri) from the data set. In general, a Richardson number below 0.25 indicates an unstable flow that initiates turbulence (Miles, 1961; Howard, 2006). Further, there is evidence that turbulence is maintained for Ri <1.0 after being initiated (e.g. Woods, 1969; Müllemann et al., 2003). Regarding potential temperatures, it is interesting to identify regions with a vertical gradient close to zero ($\frac{\partial \theta}{\partial z} \to 0$) since this indicates mixing over a specific altitude range.

During flight RF16, 15 dropsondes were launched in the near upstream region at the west edge of the mountains, above and at the east side of the mountains. Example profiles of temperature and wind measurements and the derived potential temperature and Richardson number of one dropsonde launched at 07:55 UTC (during FF05) are shown in Figure 7a. The position of the launch above the Southern Alps is marked in Figure 1 with a red dot. The thermal tropopause is found at an altitude of 10.6 km and is shown by the horizontal red dotted line in Figure 7a. In the vicinity of the thermal tropopause we find a strong vertical shear of the horizontal wind (approximately 0.02 s[-1]) induced by the tropospheric jet stream whose core region is located west of the Southern Island. In this region of vertical wind shear Ri decreases below the critical level value of 0.25 indicating dynamic instabilities and local turbulence (Pavelin et al., 2001; 2002). Simultaneously, the gradient in the potential temperature is strongly attenuated. For altitudes below 9.2 km, layers with Ri <1.0 exist which may be evidence for further





turbulence or static instabilities. In the altitude range between 9.2 and 10.2 km Ri is clearly larger than 1.0 and the potential temperature profile shows an enhanced gradient.

In Figure 7b profiles of potential temperature and Ri of two dropsondes that were launched at the same location over the Southern Alps (Figure 1) at 06:52 and 11:37 UTC show the temporal evolution over the course of the IOP. Within 5 hours the thermal tropopause descended from 11.1 km to 10.4 km. The potential temperature of the dropsonde at 11:37 UTC shows many layers with a small gradient caused by mixing processes which occurred earlier during the event. In general, the Richardson number increased in the UTLS but still shows some evidence for turbulence (Ri <1.0) right below the tropopause. At the same time the vertical shear of the horizontal winds declined (not shown) which agrees with the weakening of the gravity wave event.

The layers of suggested turbulence generally have a thickness of approximately 200 m. The gradient of the potential temperature in these layers is less than 5 K km$^{-1}$. This low gradient may be a result of initiated mixing of air masses by local turbulence.

Upstream of the mountains, wind shear regions and dynamic instabilities are not as obvious as over the middle and eastern mountains (not shown) indicating that this feature is mainly caused by the mountain waves.

## 6 Mixing identified by tracer-tracer correlation

Tracer-tracer correlations are widely used to investigate mixing of trace gases and thus can support our findings presented in the previous sections. We use the correlation between water vapor and ozone, where water vapor has a strong negative gradient in the troposphere and ozone a strong positive gradient in the lower stratosphere. In Figure 8 we show the $H_2O$-$O_3$ correlation of an unperturbed non-gravity wave Falcon flight (FF03 on 2 July 2014) and of the gravity wave flights FF04/FF05 on 4 July 2014. The flight pattern of FF03, the only flight under non-gravity wave conditions in the UTLS throughout the campaign, is similar to the gravity-wave flights, with four transects over the Southern Alps at different altitudes.

The $H_2O$-$O_3$ correlation in unperturbed conditions (grey dots) shows a clear L-shape indicating very little or no mixing of air masses. In contrast, the $H_2O$-$O_3$ correlation in the UTLS region on 4 July deviates from the L-shape. This indicates mixing in the tropopause region most likely related to the mountain waves as shown in the previous section. The mixing is strong at potential temperatures between 329 and 334 K in the UTLS region as identified by local turbulence in the dropsonde data. This potential temperature range is marked in the ozone measurements at 10.8 km altitude for flight FF04 over the middle and eastern part of the mountains (Figure 8b, inlay) where we also observed the highest mountain wave activity (section 4.3). Furthermore, we suggested in section 4.4 enhanced mixing ratios at this altitude by the shape of the vertical profile. By looking into the data in the mixing region (60 – 160 ppbv $O_3$ and 8 – 11 ppmv $H_2O$) in Figure 8b, we also find data points beyond the defined potential temperature range for mixing (329 – 334 K) (green dots). These data are located over the upstream region on the flown transect. They are not following the ideal L-shape for no mixing but have less enhanced water vapor mixing ratios than the mixed data points (red dots). The same observation is found for flight FF05 at 10.8 km altitude but for higher ozone





and lower water vapor mixing ratios since the tropopause was located at a lower altitude. Furthermore, we suggest local turbulence and induced mixing over the mountain region also for the first Falcon flight due to a similar mountain wave activity.

While the pure kinetic transport of water vapor by waves might in general be reversible, mixing implies a permanent change in the water vapor distribution in the UTLS region. The combined analysis of in-situ aircraft measurements and dropsonde data shows a transport of water vapor through the upper troposphere and lower stratosphere and a partial mixing of the air masses caused by mountain waves.

## 7 Effect on the atmospheric radiation budget

The water vapor mixing ratio in the UTLS strongly influences the radiative transfer in this region. Here, we try to derive an

estimate of the radiative forcing by the enhanced water vapor mixing ratios in the UTLS caused by the mountain waves based on simulations by Riese et al. (2012). They studied the influence of uncertainties in the atmospheric mixing strength on global UTLS distributions and the associated radiative effects of water vapor and other trace species. To this end, Riese et al. (2012) used multiannual simulations with the Chemical Lagrangian Model of the Stratosphere CLaMS (McKenna et al., 2002a; McKenna et al., 2002b). In their Figure 6, Riese et al. (2012) show the radiative effects at the top of the atmosphere of a certain

change in water vapor mixing ratios for the year 2003. For our flight conditions (approximately 300 hPa) and location (New Zealand, -45° latitude), a 10 % increase in water vapor mixing ratios near the tropopause results in a radiative forcing of 0.5 to 1 W m$^{-2}$. The percentage change between the reference and the enhanced mixing case is derived from Figure 6 in Riese et al. (2012). For our case, from the water vapor to ozone correlation we assume a minimum increase of 4 ppmv (~30 %) $H_2O$ in the mixed mountain wave region (red dots in Figure 8b) with respect to the less influenced upstream region (green dots). Under the

assumption that the change in the climatological distributions of water vapor may also be representative for our case of mixing induced by mountain waves, we estimate a radiative forcing ≥1 W m$^{-2}$ locally above New Zealand during and after the mountain wave event. An upper estimate of the radiative forcing for this case may be determined by the difference between the unperturbed conditions in flight FF03 and the mixed conditions in flights FF04/FF05. The increase in water vapor mixing ratio of ~11 ppmv (160 %) may result in a significantly larger local radiative forcing. While the analysis of Riese et al. (2012)

reflects the impact of uncertainty in the atmospheric mixing strength in the UTLS region on a global and multiannual scale, we use it here to derive a rough estimate of the local radiative effects of mountain waves for a short time period (few hours to one day). Further studies are required to evaluate the radiative forcing caused by changes in the water vapor mixing ratios due to gravity waves in more detail and/or on larger scales. However, our crude estimate shows that mountain waves have a great potential to change the water vapor distribution of the UTLS with significant effects on climate.



# 8 Conclusion and outlook

Based on in-situ aircraft measurements of water vapor and wind during the DEEPWAVE campaign we combined selected methods to investigate the vertical transport of water vapor induced by mountain waves. Flux calculations showed regions with enhanced mountain wave activity above the Southern Alps on 4 July 2014. While the meteorology of this day and the propagation of the observed mountain waves is also discussed in Bramberger (2015) and Smith et al. (2016), we concentrated on the effect of the mountain wave activity on the water vapor distribution in the UTLS. Stimulated by the flux calculation method by Shapiro (1980) and Schilling et al. (1999), we, for the first time, used in this study water vapor as a transport tracer in a wide altitude range throughout the UTLS.

Significant vertical water vapor fluxes observed by the Falcon and the GV at different flight altitudes below and above the tropopause indicated mountain wave propagation and water vapor transport through the tropopause. Forced by a strong south-westerly wind, the mountain wave activity was highest in the middle and eastern part over the Southern Alps. A wavelet analysis helped to identify the location, the direction, and the horizontal wavelength scale of the observed transport process. Covering the wavelength range of 300 m to 80 km we found an upward transport of water vapor above the mountains at horizontal wavelengths between 22 and 60 km at 8.9 km flight altitude. Further east a downward transport at smaller wavelengths <22 km occurred. Thus, the water vapor transport happened at the same horizontal wavelengths as the energy and momentum transport for this case (Smith et al., 2016).

The vertical flux divergence over the mountains within the altitude range 8.9 to 13.0 km suggests dominating upward water vapor transport through the tropopause with enhanced mixing ratios at around 10.8 km altitude and below. A downward transport in the layer between 10.8 and 12 km occurred for horizontal wavelengths <22 km and may be related to turbulence we observed in the dropsonde data. While Smith et al. (2016) and Smith & Kruse (2017) showed that there is no energy and momentum flux for these small-scale waves, we observed that a mass transport of water vapor occurred in small scales. This may point to more complex transport mechanisms of trace gases in mountain waves. To obtain the vertical water vapor flux we neglected horizontal and vertical advection in the measurement region but there were hints for additional transport processes such as convection or advection induced by the tropospheric jet stream, especially in the upstream region. These processes may also influence the measurements above the Southern Alps but they should be dominated by the vertical transport induced by the mountain waves. The occurrence of lee wave clouds at the lowest flight altitude (7.7 km) during a short time period of the first Falcon flight may additionally influence the vertical water vapor flux at 8.9 km by reducing it in the eastern part of the mountains. Since there is a time shift between the measurements at both altitudes and a vertical layer of more than 1 km between them without cloud observations, we cannot quantify the effect in this study.

In addition we investigated mixing processes induced by the mountain waves. We found indications for turbulence in dropsonde data collected over the mountain transect. Wind shear, located near and below the thermal tropopause, resulted in Richardson numbers <1.0 relevant for turbulence. We detected enhanced turbulence over few hours related to high mountain wave activity which induced mixing of water vapor in the upper troposphere over the Southern Alps. In addition the $H_2O$-$O_3$



correlation showed enhanced mixing for the mountain wave situation compared to unperturbed conditions. Thus, we explain the water vapor distribution in the UTLS for this case by a combination of vertical transport of water vapor and mixing, both related to the observed mountain waves.

The enhanced water vapor mixing ratios in the tropopause region strongly influences the radiative transfer in the UTLS. The locally and temporally limited radiative forcing over the Southern Alps exceeded 1 W m$^{-2}$ and suggests that mountain waves may have a large effect on climate.

Further studies and simulations, e.g. with the Weather Research and Forecasting (WRF) model (Wagner et al., 2017), can help to enhance our understanding of the main transport and mixing processes. For example, the influence of wind shear near the tropopause and resulting small-scale turbulence may be further investigated. Regional and global modelling could help to quantify the global changes in the UTLS water vapor distribution caused by mountain waves and their effects on the atmospheric radiation budget.

Generally, the application of our novel combination of methods to a broader data set can help to better understand the mountain wave induced change in the water vapor distribution of the UTLS and their impact on the atmospheric radiation budget.





## Acknowledgements

Part of this research was funded by the German research initiative "Role of the Middle Atmosphere in Climate (ROMIC/01LG1206A)" of the German Ministry of Research and Education in the project "Investigation of the life cycle of gravity waves (GW-LCYCLE)". Further the Deutsche Forschungsgemeinschaft (DFG) supported this work via the SFB MS-

GWaves (GW-TP/DO 1020/9-1, PACOG/RA 1400/6-1) and the HALO-SPP 1294 (grant no. VO 1504/4-1). The U.S. research was funded by NSF and NCAR/EOL. Christiane Voigt appreciates support by the Helmholtz Association under grant no. W2/W3-60. We thank the DLR flight department for excellent support of the campaign. The observational data are available at https://halo-db.pa.op.dlr.de/ and http://data.eol.ucar.edu. Michael Lichtenstern and Monika Scheibe did the ozone measurements during the campaign. Many thanks to Prof. Peter Hoor and his group from University of Mainz for the

constructive discussion about trace gas transport influenced by mountain waves. The first author also wants to thank Prof. Ron Smith for helpful hints on the data analysis and Sonja Gisinger for proofreading.

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





**Table 1.** Measurement range, accuracy and precision for the CR-2 hygrometer.

| Measurement range | Accuracy | Precision |
| --- | --- | --- |
| 50-500 ppmv | 9% | 1% |
| 10-50 ppmv | 9-12% | 2% |
| <10 ppmv | >12% | >2% |



**Table 2.** Overview of the research flights on 4 July 2014 (FF = Falcon research flight, RF = GV research flight).

| Aircraft | Flight no. | Flight time (UTC) | Dropsonde launches |
|---|---|---|---|
| DLR Falcon 20E | FF04 | 02:46 – 06:09 | -- |
| DLR Falcon 20E | FF05 | 07:23 – 11:00 | -- |
| NSF/NCAR Gulfstream V | RF16 | 05:59 – 12:55 | 15 (mountain transect); 1 (east of South Island) |





**Table 3.** Vertical flux divergence of water vapor for the combined research flights FF04, FF05 and RF16. The results are shown for two horizontal wavelength ranges.

| Flight no. | Leg number | Altitude (km) | $\frac{\partial \overline{(w'H_2O')}}{\partial z}$ (ppmv s$^{-1}$) $\lambda_h = 300\,m - 80\,km$ | $\frac{\partial \overline{(w'H_2O')}}{\partial z}$ (ppmv s$^{-1}$) $\lambda_h = 22\,km - 80\,km$ |
|---|---|---|---|---|
| FF04 | leg1→leg2 | 7.7 – 8.9 | 3.0E-02 | -2.9E-02 |
| FF04 | leg2→leg3 | 8.9 – 10.8 | -1.5E-03 | -1.1E-03 |
| FF05 | leg1→leg2 | 7.7 – 8.9 | 5.2E-02 | 4.6E-02 |
| FF05 | leg2→leg3 | 8.9 – 10.8 | -3.2E-03 | -2.2E-03 |
| FF05/RF16 | leg3 (FF05)→leg4 (RF16) | 10.8 – 12.0 | 2.5E-04 | -9.0E-04 |
| RF16 | leg4→leg5 | 12.0 – 13.0 | -1.0E-04 | -4.7E-05 |



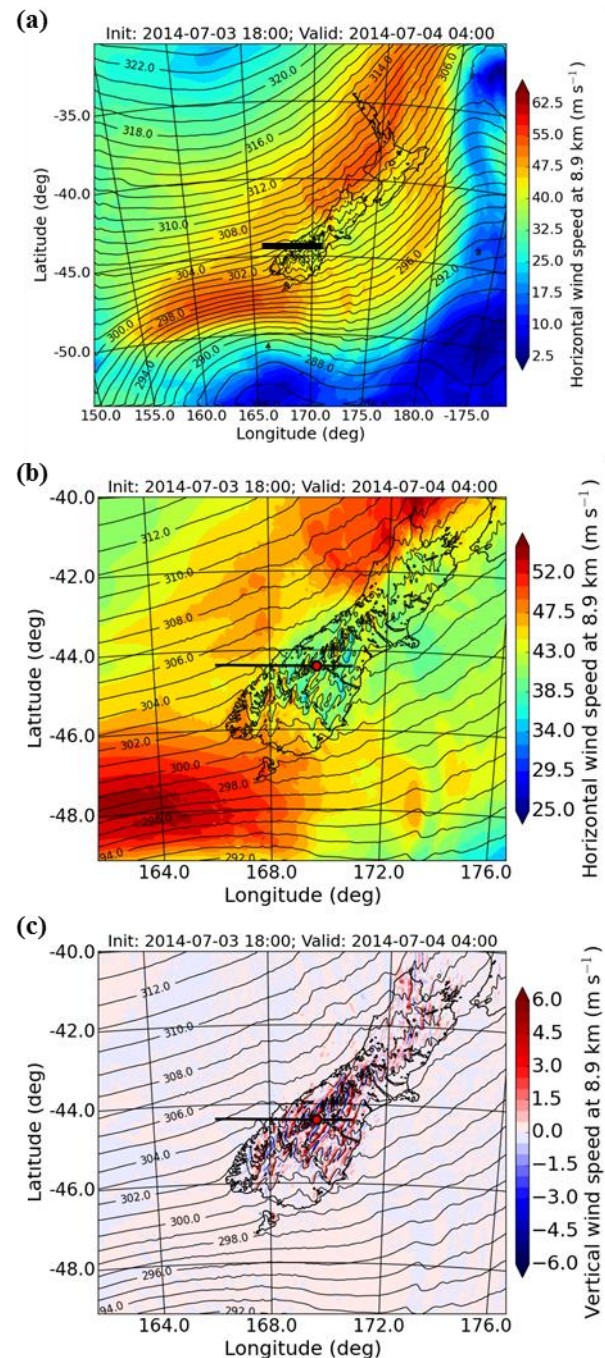

**Figure 1.** Synoptic situation on 4 July 2014 at 04 UTC: horizontal wind speed (a, b) and vertical wind speed (c) at 8.9 km (altitude of DLR Falcon flight FF04 leg 2) simulated by WRF. Contour lines represent the potential temperature at 8.9 km. The thick black line displays the cross-mountain flight path. The red dot in (b, c) marks the position of the dropsonde launch from the GV at 08 UTC (see Figure 7).





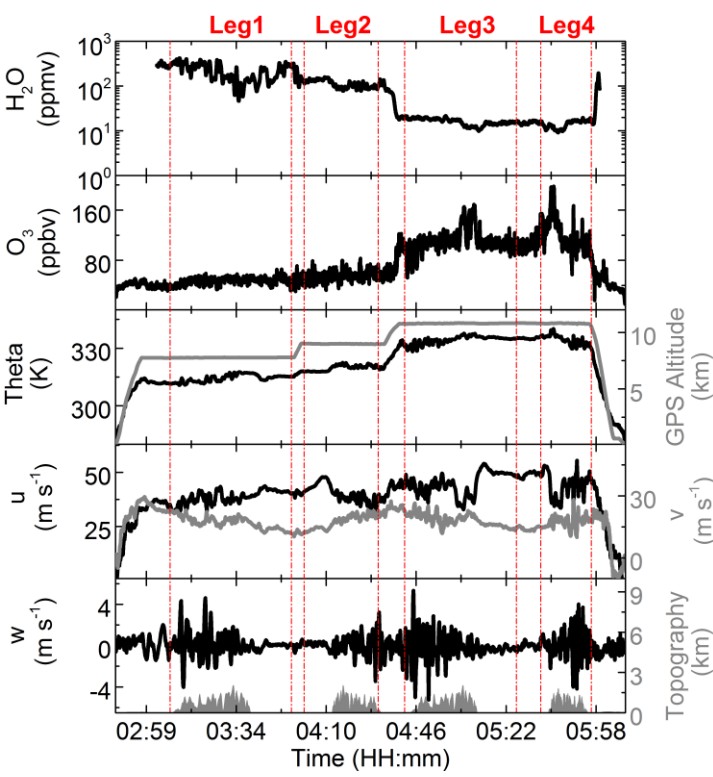

**Figure 2.** DLR Falcon flight FF04 on 4 July 2014 above the Southern Alps: time series of observations during the mountain wave event.
10   Water vapor mixing ratio (from CR-2), ozone mixing ratio, potential temperature and flight altitude as well as zonal wind, meridional wind, vertical wind and topography are shown. Flight legs are separated by dashed red lines.



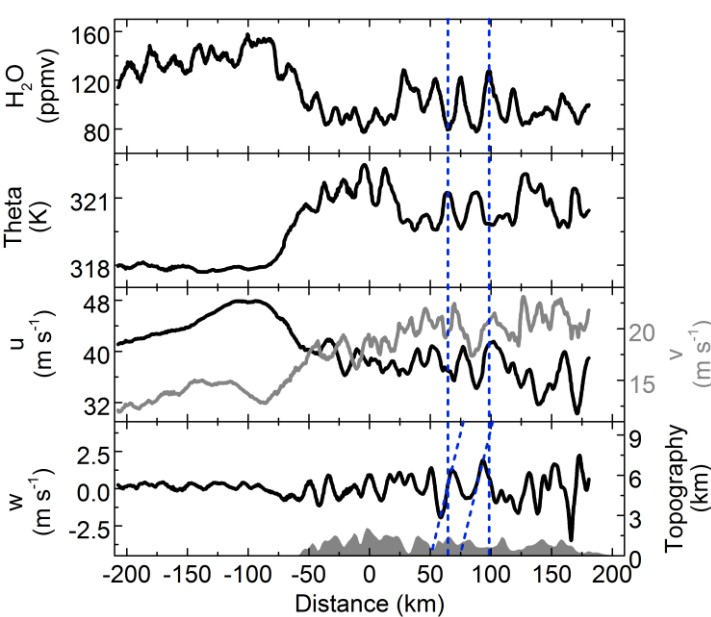

**Figure 3.** A portion of the time series of the DLR Falcon flight FF04 on 4 July 2014 shown in Figure 1. The measurements were taken during the second flight leg at 8.9 km altitude over the South Island of New Zealand. The distance refers to Mt. Aspiring as the highest summit during this mountain transect (west to east). The vertical blue dashed lines mark single wave events and the diagonal blue dashed lines in the bottom panel display the phase shift between the vertical wind motion and perturbations in water vapor and theta.





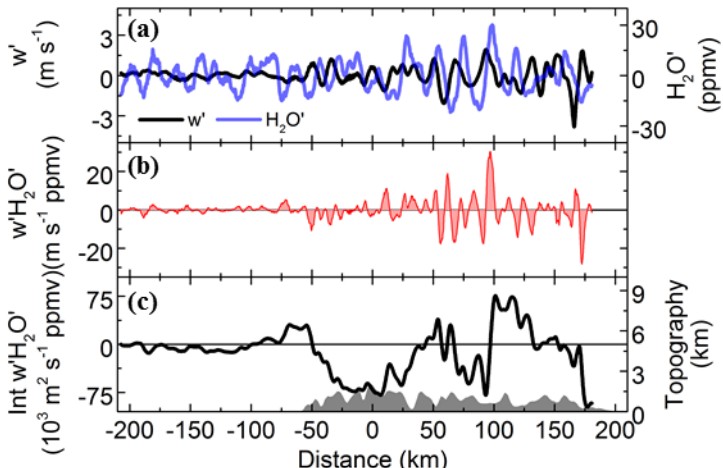

**Figure 4.** Same flight leg as in Figure 3. Shown are components of the vertical water vapor flux. The vertical wind perturbation (black) and the water vapor perturbation (blue) in (a) are combined to the local vertical water vapor flux w'$H_2O$' (b). The bottom panel (c) shows the integrated vertical water vapor flux $\int$w'$H_2O$'dx and the topography.



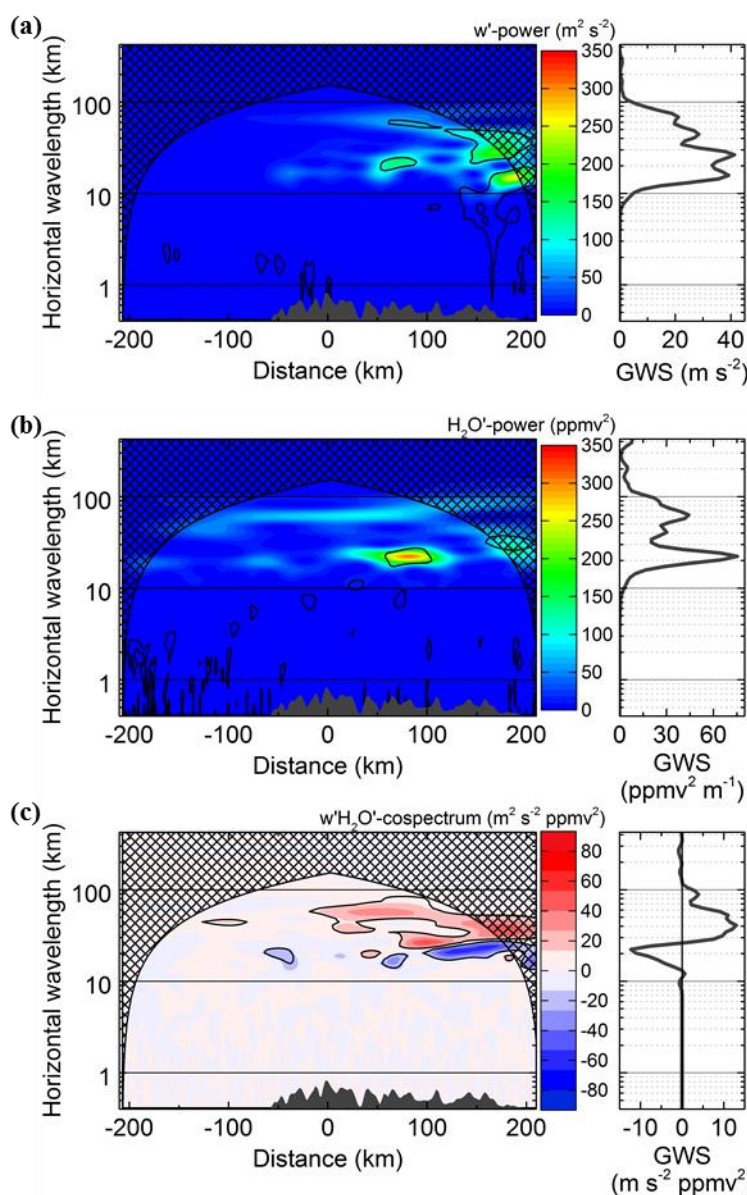

**Figure 5.** Wavelet analysis of the second flight leg of the DLR Falcon flight FF04 shown in Figure 2: (a) power spectrum of vertical wind perturbation w', (b) power spectrum of water vapor perturbation $H_2O'$, and (c) cospectrum of the vertical water vapor flux w'$H_2O'$. The right panels show the corresponding global wavelet spectrum (GWS). Thin black lines around coloured areas are the 95 % confidence level; the crosshatched area is the COI. The topography (maximum mountain height of 2049 m) is represented by the dark grey area in the bottom of each panel.





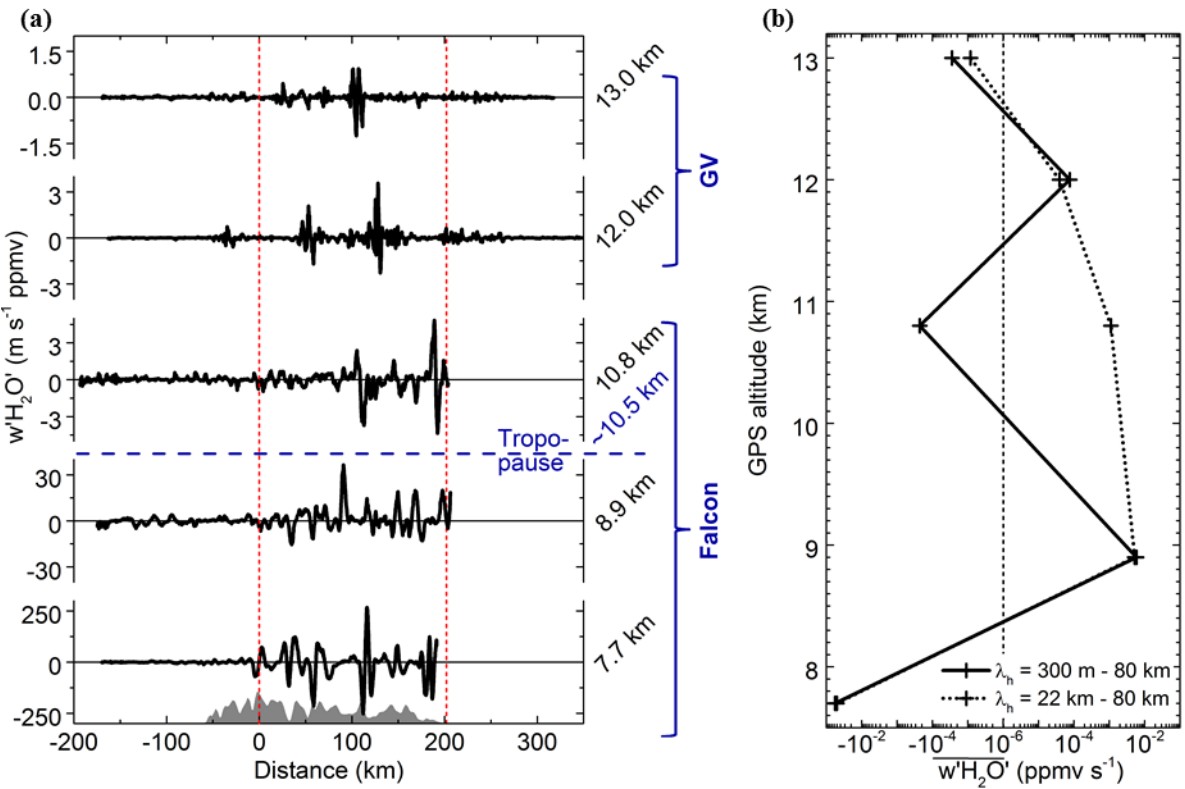

**Figure 6.** (a) Vertical water vapor fluxes using data from DLR Falcon flight FF05 (lower three panels) and NSF/NCAR GV flight RF16
(upper two panels) on 4 July 2014. The fluxes are shown for different flight altitudes over the topography of the Southern Alps. The
10  approximate height of the tropopause at 10.5 km is marked with the dashed blue line. (b) Vertical profile of the water vapor fluxes integrated
over the mountain region with highest mountain wave activity. The profiles are shown for horizontal wavelengths between 300 m and 80 km
(solid line) and between 22 km and 80 km (dashed line).

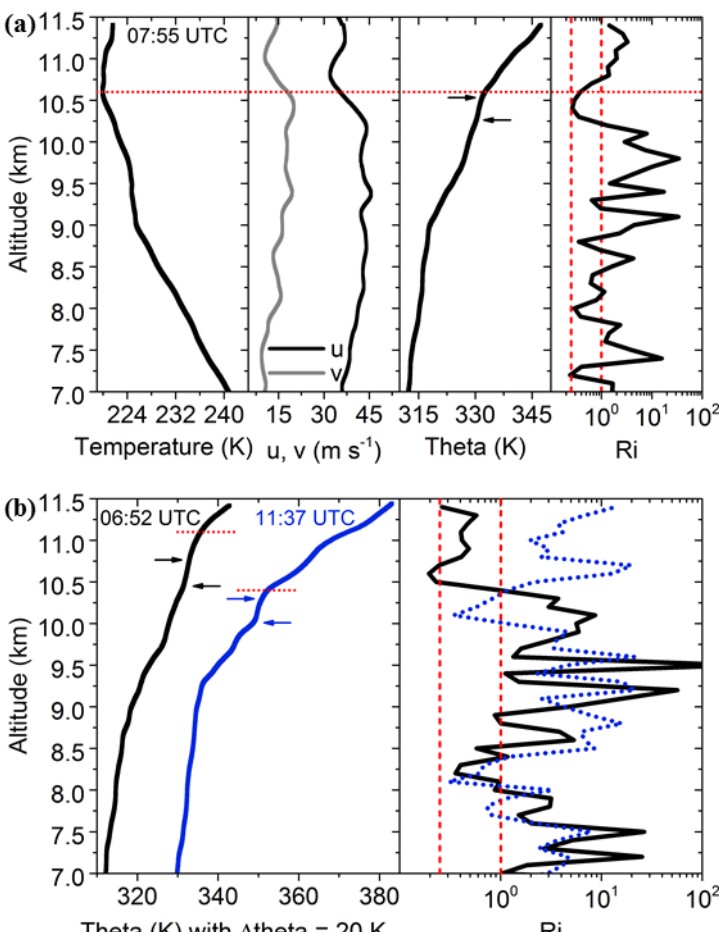

**Figure 7.** Dropsonde launches from 12.2 km height during the GV flight RF16. The panels in (a) represent the profiles of temperature, potential temperature, horizontal wind components and Richardson number for GPS altitudes from 7.0 to 11.5 km for a dropsonde at 07:55 UTC. The lower panel (b) shows the profiles of potential temperature and Richardson number for dropsondes launched at the same location as the dropsonde from (a) at 06:52 UTC (black) and 11:37 UTC (blue). The red dashed lines in the right panel show critical Ri at 0.25 and 1.0, the arrows in the theta panel denote regions with suggested turbulence. Horizontal red dotted lines in (a) and (b) mark the height of the thermal tropopause.





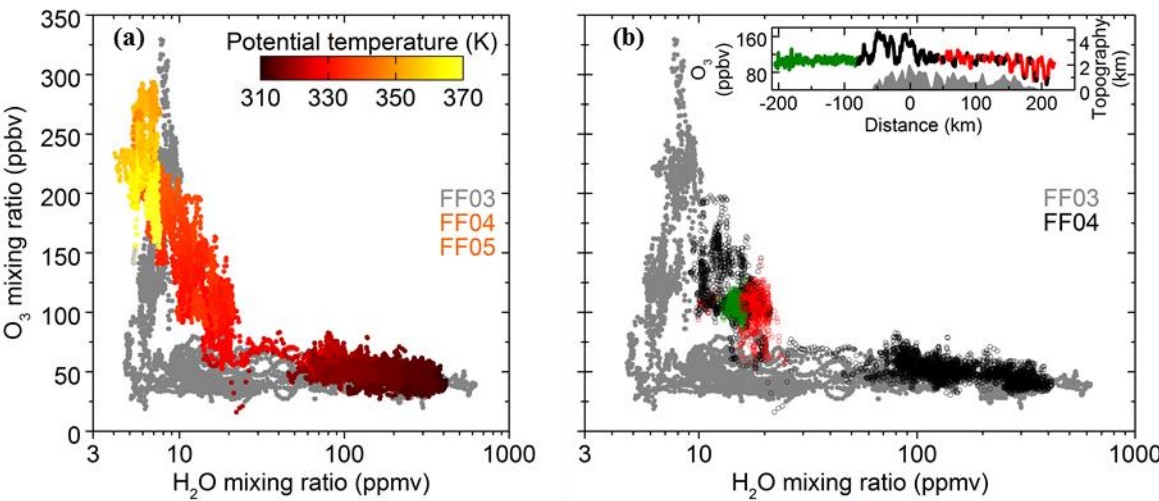

**Figure 8.** $H_2O$-$O_3$ correlation for three Falcon flights: (a) FF04 and FF05 in mountain wave conditions and FF03 in unperturbed conditions. Potential temperature is colour-coded for FF04 and FF05. (b) FF04 with a red-marked region for potential temperatures between 329 and 334 K that correspond to regions where turbulence in the dropsonde data of flight RF16 was observed. The inlay in (b) gives the ozone mixing ratio of flight FF04 leg 3 at 10.8 km. The red data points show the localisation of potential temperatures between 329 and 334 K in
15  the ozone data and green data points mark the upstream region of this flight leg.

