# Peer review of "Mountain waves modulate the water vapor distribution in the UTLS"

_Atmospheric Chemistry and Physics, 2017_

## Referee Comment (RC1) · Anonymous Referee #1 · 24 May 2017

The authors present an interesting case study addressing vertical transport and irreversible mixing of water vapor in the UTLS associated with a mountain wave event. The wave event located above the Southern Alps of New Zealand is analysed involving observations from two aircrafts and radiosondes. A novel combination vertical flux calculations, wavelet analyses and tracer-tracer correlations is used to identify regions of vertical trace gas fluxes, involved horizontal wavelengths, and indications for irreversible mixing. The analysed data set suggests a dominating upward water vapor transport through the thermal tropopause followed by partial mixing, thus enhancing water vapor mixing ratios in the tropopause region. Furthermore, based on simulations by Riese et al. (2012), a rough estimate of a local $\geq 1$ W/m$^2$ radiative forcing due to mountain wave-induced water vapor transport and irreversible mixing is provided.

The paper is clearly of interest, as trace gas and particularly water vapor transport

by mountain waves followed by irreversible mixing is little understood and represents a source of uncertainty in simulations. The presented set of observations supports a consistent picture of local upward transport by mountain waves and partial mixing, resulting in a net enhancement of water vapor in the tropopause region. The estimated radiative forcing should be taken with care, since local observations at a certain time of the year are combined with zonally and temporally averaged data. As indicated by the authors, this aspect clearly requires further studies. The paper is clear-written and well structured. The study should be published in ACP after clarification of some minor points:

P 5 line 16 and elsewhere: cloud-free conditions, water as conservative tracer: As potential condensation may influence the analysis, the absence of (thin) clouds should be assured using airborne data (e.g. particle observations or temperature). Later it is said that ice particles were detected at the leewave side. What is the detection limit for condensed water? Could significant amounts of condensed water be missed, or can this be ruled out?

P 5 line 28: To me it was sometimes difficult to connect the flight legs and locations/directions with the map in Fig. 1. As Mt. Aspiring serves as reference point, coordinates should be provided in the text and it would be helpful to mark this point in the maps.

P8 sect. 4.1 and Figure 1: As the vertical domain is in the focus of this study and locations are relevant, it may be helpful to add a vertical cross section of vertical wind from the model along the cross-mountain flight path and indicate the flight legs.

P12 line 4, Figure 6: While the data suggest upward transport through the thermal tropopause, it would be interesting to include comment on the dynamical tropopause. Are thermal and dynamical tropopause approximately coincident here? Furthermore, how is the approximate thermal tropopause location determined in Figure 6 (dropsondes/model)? Could the location be biased by temperature signatures of the strong

waves?

P12 line 6, Figure 6b: exact localization of maxima ... not possible: It is clear that it is difficult to have observations at many different levels in a short time window and here the best possible is done. However, could the pattern in Figure 6b change significantly if more/other levels would be available?

P13 line 25: Turbulence is identified in the dropsonde data between 329 and 334 K and suggests mixing. Figure 7b shows that the situation is changing within hours. Is it robust to apply this potential temperature range from a single dropsonde profile to the H2O-O3 correlation from a full flight covering several hours?

P13 line 21: A local $\geq 1$ W/m$^2$ radiative forcing is estimated locally above New Zealand in July. However, Figure 6 in Riese et al. (2012) refers to annually and zonally averaged values. How could this affect this estimate?

Technical:

P3 line 3: correlations

P6 Eqns 1 and 2: define x and t

P9 line 18: check number/unit: -176 m ppmv

P9 line 28: strong negative peak

Figure 5: numbers at right y-axes of panels on the right side would be helpful

---

## Referee Comment (RC2) · Anonymous Referee #2 · 13 Jun 2017

This study addresses the modulation of water vapour in the upper troposphere/lower stratosphere by mountain waves. It draws on a wealth of aircraft measurements made over New Zealand in the context of the DEEPWAVE campaign, and puts them to good use, combined with numerical simulations and soundings. The paper contains a rather thorough processing of these data (for example, using wavelet analysis), with the aim of understanding how mountain waves influence the behaviour of atmospheric water vapour near the tropopause. The work is highly relevant scientifically, namely because it reports on novel data, and may have climate implications, and is suitable for the scope of ACP. Both previous work on the topic and the scientific approach and methods are adequate and discussed in appropriate detail. The number of figures, tables and references included also seems appropriate. The conclusions presented are interesting,

relevant and supported by the results. The manuscript is well organized and written in good-quality, clear English.

General comments

Since, as pointed out by the authors, the fluctuations of water vapour in an atmosphere with strong gradients of this substance can be explained using a mixing-length argument, it would be nice to see how well the mixing length obtained from this kind of argument (i.e. defined as the magnitude of the water vapour fluctuations divided by the water vapour gradient) compares with the wave amplitude obtained directly from integrating the vertical velocity. This would, presumably, give indications about the mixing effectiveness, as a mixing length substantially smaller than the diagnosed wave amplitude would suggest considerable fluid parcel dilution.

In Section 5 and Figure 7, some attention is devoted to the vertical profiles of the potential temperature theta and the wind velocity (U, V), for the purpose of calculating the Richardson number Ri. Although this is obviously highly relevant from the standpoint of turbulence generation, it would also be interesting to add panels to Figure 7 containing Scorer parameter profiles, computed from the same quantities, and discuss the implications of the vertical structure of these profiles in terms of vertical propagation (or trapping) and amplification (or decay) of the mountain waves.

Specific comments

Page 2, Lines 23-24: "The transport of trace gas species may be reversible or irreversible, depending on mixing processes on different scales.". This sentence as it stands could be misleading. Any mixing will cause irreversibility, yet the reader gets the impression that reversibility depends on the scale at which mixing occurs. Consider rephrasing to clarify.

Page 3, lines 3-4: "The tracer-tracer correlation are based on a dynamic approach". Please replace "correlation" with "correlations". What is meant by "dynamic approach"

here? Is the purpose simply making a contrast with "microphysics" mentioned later in the sentence? If yes, this should be better explained.

Page 7, line 20: "with a lag of one and a lag of 10". It is not obvious to the reader why these values are used. Perhaps the authors should cite here (again) the reference where these assumptions are motivated.

Page 11, lines 6-9: "In their study flux-carrying waves are larger than 20km horizontal wavelength. Small scale waves with wavelengths around 20km and less are mainly dominating in the vertical wind motion and do not carry any energy or momentum flux upward". It should be noted that, in the case of momentum or energy, the reason for this behaviour is dynamical, since only large-scale waves that propagate vertically (i.e. are not evanescent) transport momentum and energy vertically. For water vapour, this scale filtering cannot occur for the same reasons, since water vapour may be viewed as an essentially passive tracer.

Page 14, lines 19-22: "Under the assumption that the change in the climatological distribution of water vapour may also be representative for our case of mixing induced by mountain waves, we estimate a radiative forcing > 1 W m$^{-2}$ locally above New Zealand during and after the mountain wave event.". It would be good to discuss the validity of this assumption a bit further. Under what circumstances is it expected to fail?

Page 14, lines 27-28: "Further studies are required to evaluate the radiative forcing caused by changes in the water vapor mixing ratios due to gravity waves in more detail and/or on larger scales.". Why specifically on larger scales? What scales in particular?

Page 16, lines 4-6: "The locally and temporally limited radiative forcing over the Southern Alps exceeded 1 W m$^{-2}$ and suggests that mountain waves may have a large effect on climate.". I suspect this may be an overstatement. To ascertain whether this claim is reasonable, the prevalence of mountain waves similar to those addressed in the present study would have to be taken into account. The tone of this remark could be moderated.

Page 28, Figure 4: I do not think the large negative flux of water vapour that can be seen in the bottom graph between x=-50 km and x=+50 km is discussed in sufficient detail in the text. This is an intriguing feature, which may seem puzzling to the reader. I advise the authors to include an interpretation of it, even if speculative, justifying its intensity, location and extent.

Page 30, Figure 6: In panel(a), the caption does not explain what the red dashed lines represent. Please add that information. In panel (b), the dotted line corresponding to the water vapour flux filtered for waves with wavelengths between 20 km and 80 km does not include a point at z=7.7 km, but the solid line does. Why is that? This choice should be justified convincingly in the text.

Technical corrections

Page 1, Line 13 in Abstract: "GV research aircraft". Does "GV" stand for anything, or is it just the name of the aircraft? In the first possibility, please expand and explain the acronym.

Page 7, line 9: "q=H_2 O". If the two notations are equivalent, why use "H_2 O" instead of the shorter and more convenient notation "q", as is done throughout the manuscript? Is there any particular reason for this?

Page 11, lines 29-30: "By the absence of vertical or horizontal transport and the existence of a well-mixed atmosphere, we are expecting no flux divergence.". This sentence does not sound very well. Consider rephrasing.

Page 25, Figure 1(b): Is this simply a magnification of Figure 1(a)? If yes, this should be mentioned in the caption.

Page 26, caption of Figure 2: "... and topography are shown". It would be good to indicate what denotes the topography, i.e. the grey area at the bottom (as done in e.g. Figure 5).

Page 27, caption of Figure 3: "the diagonal blue dashed lines in the bottom panel display the phase shift between the vertical wind motion and perturbations in water vapor and theta". It is not totally clear what this means. Does the phase shift correspond to the horizontal distance between the bottom and top of these lines? Please clarify.

---

## Author Comment (AC1) · 6 Oct 2017

Response to Anonymous Referee #1 (acp-2017-334-RC1)

*Our responses are written in italic.*

*The changes in the manuscript are transferred and marked as quotations.*

*We thank the reviewer for his/her positive assessment of the manuscript and the helpful comments.*

The authors present an interesting case study addressing vertical transport and irreversible mixing of water vapor in the UTLS associated with a mountain wave event …

The paper is clearly of interest, as trace gas and particularly water vapor transport by mountain waves followed by irreversible mixing is little understood and represents a source of uncertainty in simulations. The presented set of observations supports a consistent picture of local upward transport by mountain waves and partial mixing, resulting in a net enhancement of water vapor in the tropopause region. The estimated radiative forcing should be taken with care, since local observations at a certain time of the year are combined with zonally and temporally averaged data. As indicated by the authors, this aspect clearly requires further studies. The paper is clear-written and well structured. The study should be published in ACP after clarification of some minor points:

P 5 line 16 and elsewhere: cloud-free conditions, water as conservative tracer: As potential condensation may influence the analysis, the absence of (thin) clouds should be assured using airborne data (e.g. particle observations or temperature). Later it is said that ice particles were detected at the leewave side. What is the detection limit for condensed water? Could significant amounts of condensed water be missed, or can this be ruled out?

*We detected cirrus clouds during the campaign by measuring total water with a tunable diode laser hygrometer with a forward-facing inlet. We calculated the ice water content by subtracting the saturation mixing ratio from the total water signal and correction for particle enhancement. The resulting detection limit for the ice water content is 0.2 ppmv. The clouds in the lee of the New Zealand Alps that are mentioned in the manuscript had an ice water content of 10 to 200 ppmv and were observed on the lowest flight leg (7.7 km) of the first Falcon flight. This is in agreement with satellite measurements. No clouds were observed at higher flight altitudes of this flight and during the second Falcon flight. The possible influence of these clouds is discussed in the manuscript.*

"For the flux calculations we used water vapor as conservative tracer due to the absence of supersaturation at the analysed flight altitudes. However, at the first flight leg of FF04 at 7.7 km we measured ice particles with the in-situ instrumentation with a detection limit for the ice water content of 0.2 ppmv. The cloud was detected between +150 km and +200 km distance and indicates the existence of a lee wave cirrus. This gravity wave induced cloud was also visible in the infra-red images of the MTSAT-2 satellite at 03 UTC and dissipated until 06 UTC (Bramberger et al., 2017). No further clouds were measured on the other flight legs and in particular not during those legs for which the flux calculations were performed. However, the presence of an ice cloud on a lower layer may affect the water vapor distribution at a higher flight level (8.9 km) by lowering the amplitude of the fluctuation. In Figure 3 we observe a strong negative peak in the vertical wind at +170 km distance to the summit in contrast to a small water vapor fluctuation which may be influenced by the drying of the level below. The calculated flux in this region is then also reduced. This effect does not influence the general transport direction at this flight altitude and is not relevant for the higher flight altitudes or the second Falcon flight since these lee wave clouds were not observed above 7.7 km and dissipated during the first flight."

P 5 line 28: To me it was sometimes difficult to connect the flight legs and locations/ directions with the map in Fig. 1. As Mt. Aspiring serves as reference point, coordinates should be provided in the text and it would be helpful to mark this point in the maps.

*In the revised manuscript we have marked Mt. Aspiring in Fig. 1 and have provided the coordinates.*

P8 sect. 4.1 and Figure 1: As the vertical domain is in the focus of this study and locations are relevant, it may be helpful to add a vertical cross section of vertical wind from the model along the cross-mountain flight path and indicate the flight legs.

*We now have added a vertical cross section of the vertical wind in Fig. 1 and have marked the flight leg for which model calculations were carried out for better clarity.*

P12 line 4, Figure 6: While the data suggest upward transport through the thermal tropopause, it would be interesting to include comment on the dynamical tropopause. Are thermal and dynamical tropopause approximately coincident here? Furthermore, how is the approximate thermal tropopause location determined in Figure 6 (dropsondes/model)? Could the location be biased by temperature signatures of the strong waves?

*The height of the thermal tropopause on 4 July 2014 was determined by output of numerical weather prediction models (ECMWF and WRF). These findings were confirmed by analyzing the dropsonde temperature profiles. The dynamical tropopause was derived as the 2-PVU level in the WRF model data. It is approximately coincident with the thermal tropopause, i.e., it is just located a few hundred meters below. In the region of the observed mountain waves the tropopause (thermal as well as dynamical) is more structured and potentially biased by wave signatures than up- or downstream the mountains. Therefore, the given heights of the tropopause are averages over the whole flight leg distance.*

P12 line 6, Figure 6b: exact localization of maxima : : : not possible: It is clear that it is difficult to have observations at many different levels in a short time window and here the best possible is done. However, could the pattern in Figure 6b change significantly if more/other levels would be available?

*We added a short note on this discussion topic. The first Falcon flight shows the same pattern of vertical divergence for the lower altitudes. A significant change should not appear by using other levels, but maxima could possibly shift in altitude. In another field campaign conducted two years after DEEPWAVE, a lidar for water vapor measurements pointed upward and made it possible to derive a profile of the vertical water vapor flux. However, the operating lidar instruments on board the Falcon and the GV were not measuring water vapor during DEEPWAVE.*

"For the first Falcon flight we find a similar pattern and values for the flux divergence between 7.7 and 10.8 km (Table 3). The use of other levels could change the pattern slightly but the general trend appears to be robust. Vertically resolved data (e.g. by lidar measurements) would be required to derive the vertical curtain of the flux divergence but were not performed during this campaign."

P13 line 25: Turbulence is identified in the dropsonde data between 329 and 334 K and suggests mixing. Figure 7b shows that the situation is changing within hours. Is it robust to apply this potential temperature range from a single dropsonde profile to the $H_2O$-$O_3$ correlation from a full flight covering several hours?

*We clarify this remark. The potential temperature range of 329 to 334 K for turbulence is derived from a total of 9 dropsonde launches during the GV flight. These dropsondes were launched over 5 hours so we assume a similar range for the first Falcon flight. The altitude range of the turbulence occurence changed due to a change in the tropopause height.*

"The layers of suggested turbulence, found in all nine dropsondes launched above the middle and eastern part of the mountains, generally have a thickness of approximately 200 m and are correlated with a potential temperature range of 329 to 334 K."

"The dropsondes, covering a time range of 5 hours before, during and after the second Falcon flight FF05, always show turbulence in the same potential temperature range with slight changes in the altitude due to the descent of the thermal tropopause. Thus, we also assume the presence of turbulence layers for similar potential temperatures during the first Falcon flight FF04."

P13 line 21: A local _ 1 W/m2 radiative forcing is estimated locally above New Zealand in July. However, Figure 6 in Riese et al. (2012) refers to annually and zonally averaged values. How could this affect this estimate?

*We have tried to moderate our statement by emphasizing the difference between our own study and the study of Riese et al. (2012). We attempted to relate the results of Riese et al. (2012) to our measurement region and then derived a lower limit for the radiative effect. Since we compare our case for mixing by mountain waves with annually and zonally averaged values a large uncertainty exist. Water vapor shows a seasonality in the mid latitude region of the Southern Hemisphere (e.g. Hegglin et al., 2013[1]) with lower mixing ratios in July than in January. The use of absolute values for the difference in the water vapor mixing ratio induced by mixing would lead to a biased estimation of the radiative forcing. However, we based our estimation on the percentage difference which slightly reduces the uncertainty for a comparison of annually averaged mixing ratios with a case on a specific day.*

"Under the assumption that the simulated difference in the distribution of water vapor as a result of enhanced mixing may also be representative for our case of mixing induced by mountain waves, we estimate a radiative forcing larger than 1 W m$^{-2}$ locally above New Zealand during and after the mountain wave event. Riese et al. (2012) do not give a physical reason for the changes in the mixing strength, so our case may present a physical process (among other processes) contributing to the change in the water vapor distribution in the UTLS. While we used the calculations by Riese et al. (2012) at the measurement location, their study has a coarser vertical and horizontal resolution and is averaged over one year. We here neglect the seasonality in the water vapor mixing ratio that is present in the southern hemisphere at this latitude range. Thus, our estimate has a large uncertainty. Nevertheless, it emphasizes the relevance of mountain waves on the water vapor distribution and the radiation budget of the UTLS."

Technical:

*Thank you for the following remarks; they are now corrected in the manuscript.*

P3 line 3: correlations

P6 Eqns 1 and 2: define x and t
* * *
[1] Hegglin, M. I., Tegtmeier, S., Anderson, J., Froidevaux, L., Fuller, R., Funke, B., Jones, A., Lingenfelser, G., Lumpe, J., Pendlebury, D., Remsberg, E., Rozanov, A., Toohey, M., Urban, J., von Clarmann, T., Walker, K. A., Wang, R., and Weigel, K.: SPARC Data Initiative: Comparison of water vapor climatologies from international satellite limb sounders, J. Geophys. Res. Atmos., 118(20), 11,824-811,846, doi:10.1002/jgrd.50752, 2013.

P9 line 18: check number/unit: -176 m ppmv

P9 line 28: strong negative peak

Figure 5: numbers at right y-axes of panels on the right side would be helpful

---

## Author Comment (AC2) · 6 Oct 2017

Response to Anonymous Referee #2 (acp-2017-334-RC2)

*Our responses are written in italic.*

*The changes in the manuscript are transferred and marked as quotations.*

*We thank the reviewer for his/her helpful comments and suggestions to improve the manuscript.*

This study addresses the modulation of water vapour in the upper troposphere/lower stratosphere by mountain waves. It draws on a wealth of aircraft measurements made over New Zealand in the context of the DEEPWAVE campaign, and puts them to good use, combined with numerical simulations and soundings. The paper contains a rather thorough processing of these data (for example, using wavelet analysis), with the aim of understanding how mountain waves influence the behaviour of atmospheric water vapour near the tropopause. The work is highly relevant scientifically, namely because it reports on novel data, and may have climate implications, and is suitable for the scope of ACP. Both previous work on the topic and the scientific approach and methods are adequate and discussed in appropriate detail. The number of figures, tables and references included also seems appropriate. The conclusions presented are interesting, relevant and supported by the results. The manuscript is well organized and written in good-quality, clear English.

General comments

Since, as pointed out by the authors, the fluctuations of water vapour in an atmosphere with strong gradients of this substance can be explained using a mixing-length argument, it would be nice to see how well the mixing length obtained from this kind of argument (i.e. defined as the magnitude of the water vapour fluctuations divided by the water vapour gradient) compares with the wave amplitude obtained directly from integrating the vertical velocity. This would, presumably, give indications about the mixing effectiveness, as a mixing length substantially smaller than the diagnosed wave amplitude would suggest considerable fluid parcel dilution.

*We fully agree with the reviewer that a calculation of the mixing length and comparison to the wave amplitude would be useful to evaluate the mixing effectiveness. However, the calculation of a mixing length is not possible since we do not have a realistic water vapor profile over the mountains. During the analysis of this case study we thought about calculating the vertical exchange coefficient $K_q$ ($K_q = -\frac{\overline{w'q'}}{\frac{d\bar{q}}{dz}}$) that is similar to the mixing length argument you mentioned ($\frac{\overline{q'}}{\frac{d\bar{q}}{dz}}$) but for both methods a vertical gradient of $H_2O$ is necessary. From our in-situ measurements of water vapor on the Falcon and on the GV we derived an approximate profile from ascent and descent and from changes between the flight levels upstream or downstream the mountains. In the region of the observed mountain waves no vertical water vapor profile exists. Also, we cannot use the water vapor profile from the dropsondes since they were launched at 12.2 km altitude and the humidity sensor does not work properly 2 to 3 km below the launch altitude (very low humidity in tropopause region). A vertical profile of water vapor from model simulations (ECMWF, WRF) is also not applicable because the humidity of the models and the in-situ measurement in the tropopause region shows discrepancies in absolute values and in the amplitude of the fluctuations. A detailed comparison would be necessary. Additionally, we tested a method to determine the vertical gradient of water vapor from a correlation between the vertical displacement and the water vapor fluctuations on each flight leg. The method was adapted from Smith et al. (2008)[1] displayed in their*
* * *
[1] Smith, R. B., Woods, B. K., Jensen, J., Cooper, W. A., Doyle, J. D., Jiang, Q. F., and Grubisic, V.: Mountain waves entering the stratosphere, J. Atmos. Sci., 65(8), 2543-2562, doi:10.1175/2007jas2598.1, 2008.

*Fig. 9 and eq. 3. The vertical gradient of water vapor can be estimated from the slope of the linear relation between vertical displacement and water vapor fluctuations. However, the correlation scattered too much to derive a linear relationship between the two parameters. Thus, a vertical gradient of water vapor in the sensitive tropopause region has a large uncertainty. Due to this fact, we decided to leave out the discussion on the mixing effectiveness from the manuscript.*

In Section 5 and Figure 7, some attention is devoted to the vertical profiles of the potential temperature theta and the wind velocity (U, V), for the purpose of calculating the Richardson number Ri. Although this is obviously highly relevant from the standpoint of turbulence generation, it would also be interesting to add panels to Figure 7 containing Scorer parameter profiles, computed from the same quantities, and discuss the implications of the vertical structure of these profiles in terms of vertical propagation (or trapping) and amplification (or decay) of the mountain waves.

*We are grateful for this comment that gives an additional and interesting statement to our manuscript by explaining the observed wavelengths that are responsible for a vertical water vapor flux. The profile of the Scorer parameter is similar for all dropsondes launched during the flight RF16. It shows that linear gravity waves with horizontal wavelengths larger than 10 to 20 km can vertically propagate in the troposphere if they are excited by the flow over the mountain. In the middle and upper troposphere, the critical wavelength increases slightly resulting in an evanescent behavior for horizontal wavelengths smaller than about 22 km. These results of the vertical dependence of the Scorer parameter confirm our observations of horizontal wavelengths larger than 20 km that transport water vapor upward or downward.*

"Another characteristic factor is the Scorer parameter ℓ that is shown in Figure 7c for the dropsonde launched at 07:55 UTC (44.39°S, 169.60°E) The Scorer parameter is used to estimate the critical horizontal wavelengths allowing vertical propagation of linear gravity waves under the given atmospheric conditions. The vertical profile of ℓ shows that gravity waves with horizontal wavelengths between 10 and 20 km are able to propagate vertically if they are excited in the lower troposphere. Between 4 and 9 km altitude, wave modes with horizontal wavelengths smaller than the critical wavelength of about 22 km become evanescent and may be attenuated. The magnitude of the estimated critical wavelength based on the Scorer parameter confirms our observations in the power spectra and wavelet cospectrum (Figure 5): the upward transport of water vapor is dominated by horizontal wavelengths larger than 22 km. A downward transport is possible by wavelengths smaller than 22 km due to a wave attenuation in the upper troposphere that is responsible for damping and partial reflecting of gravity waves. The vertical profile of ℓ is similar for all dropsonde launches (upstream and over the mountains) and is also comparable to an upstream ℓ-profile from the IFS forecast shown in Figure 3b in Bramberger et al. (2017)."

Specific comments

Page 2, Lines 23-24: "The transport of trace gas species may be reversible or irreversible, depending on mixing processes on different scales.". This sentence as it stands could be misleading. Any mixing will cause irreversibility, yet the reader gets the impression that reversibility depends on the scale at which mixing occurs. Consider rephrasing to clarify.

*We agree and have rephrased the sentence to clarify the issue of (ir)reversibility of transport processes.*

Page 3, lines 3-4: "The tracer-tracer correlation are based on a dynamic approach". Please replace "correlation" with "correlations". What is meant by "dynamic approach here? Is the purpose simply making a contrast with "microphysics" mentioned later in the sentence? If yes, this should be better explained.

*We refer to dynamic transport processes as the main reason for the behavior of the tracer-tracer correlations and make clear that water vapor is special due to additional microphysical processes (cloud formation and freezing out of water vapor).*

"In addition to transport and mixing processes, in cloudy situations, the tracer-tracer correlations for water vapor may additionally be affected by microphysical processes and cloud formation. Then, effects of clouds on the correlations have to be discussed in such situations."

Page 7, line 20: "with a lag of one and a lag of 10". It is not obvious to the reader why these values are used. Perhaps the authors should cite here (again) the reference where these assumptions are motivated.

*The used autocorrelation factor in eq. 9 is determined by the method of Portele et al. (2017) in the way that large and small wavelengths are weighted similarly to be significant.*

"The original time series is correlated with a delayed copy of itself (time lag) to obtain the significant parts of the cospectrum. The chosen combination includes signals of larger wavelengths (significant for high time lags) and smaller wavelengths (significant for lower lags) without stressing any of them (Portele et al., 2017)."

Page 11, lines 6-9: "In their study flux-carrying waves are larger than 20km horizontal wavelength. Small scale waves with wavelengths around 20km and less are mainly dominating in the vertical wind motion and do not carry any energy or momentum flux upward". It should be noted that, in the case of momentum or energy, the reason for this behaviour is dynamical, since only large-scale waves that propagate vertically (i.e. are not evanescent) transport momentum and energy vertically. For water vapour, this scale filtering cannot occur for the same reasons, since water vapour may be viewed as an essentially passive tracer.

*We thank for this remark and agree that there are dynamical reasons for the behavior of the energy and momentum flux. Nevertheless, for this wavelength scale range (20 – 80 km) water vapor is a passive tracer for this wave dynamics. Since we use similar equation for the flux calculations the scale separation for momentum and energy transport is comparable to that for the trace gas transport. For larger scales (>100 km) other processes would be dominant but this is not in the scope of this analysis.*

"In the statistical analysis of all GV flight level data during DEEPWAVE, Smith et al. (2016) also observed small and longer scale waves with different characteristics. In their study flux-carrying waves are larger than 20 km horizontal wavelength. Small scale waves with wavelengths around 20 km and less are mainly dominating in the vertical wind motion and do not carry any energy or momentum flux upward (Smith & Kruse, 2017). This is explained by dynamic reasons since only the longer-scale waves that propagate vertically and are not evanescent transport energy and momentum vertically. For water vapor as passive tracer the reasons for the chosen scale separation are the same in this wavelength range. Transport processes by large-scale waves with horizontal wavelengths larger than 100 km would be presumably different for energy or momentum and water vapor."

Page 14, lines 19-22: "Under the assumption that the change in the climatological distribution of water vapour may also be representative for our case of mixing induced by mountain waves, we estimate a radiative forcing > 1 W m^-2 locally above New Zealand during and after the mountain wave event.". It would be good to discuss the validity of this assumption a bit further. Under what circumstances is it expected to fail?

*We discuss our assumption in more detail and emphasize the difference between our case and the study of Riese et al. (2012). For our estimation we use the same latitude range and altitude. Nevertheless, a large uncertainty exists since Riese et al. (2012) has a coarser resolution in vertical and horizontal dimension and is based on*

*zonally and annually averaged values. Additionally, our assumption is expected to fail in the absence of permanent mixing.*

"Under the assumption that the simulated difference in the distribution of water vapor as a result of enhanced mixing may also be representative for our case of mixing induced by mountain waves, we estimate a radiative forcing larger than 1 W m$^{-2}$ locally above New Zealand during and after the mountain wave event. Riese et al. (2012) do not give a physical reason for the changes in the mixing strength, so our case may present a physical process (among other processes) contributing to the change in the water vapor distribution in the UTLS. While we used the calculations by Riese et al. (2012) at the measurement location, their study has a coarser vertical and horizontal resolution and is averaged over one year. We here neglect the seasonality in the water vapor mixing ratio that is present in the southern hemisphere at this latitude range. Thus, our estimate has a large uncertainty. Nevertheless, it emphasizes the relevance of mountain waves on the water vapor distribution and the radiation budget of the UTLS."

Page 14, lines 27-28: "Further studies are required to evaluate the radiative forcing caused by changes in the water vapor mixing ratios due to gravity waves in more detail and/or on larger scales.". Why specifically on larger scales? What scales in particular?

*We have removed the remark on larger scales (horizontal wavelengths larger than 100 km). The flux calculations refer to horizontal wavelengths smaller than 80 km but in the $H_2O$-$O_3$ correlation all wavelength scales are included. Therefore, our estimation on the radiative forcing covers all wavelength scales. A separation of the impact of different wavelength scales on the radiative forcing would be difficult and is beyond the scope of this study.*

Page 16, lines 4-6: "The locally and temporally limited radiative forcing over the Southern Alps exceeded 1 W m^-2 and suggests that mountain waves may have a large effect on climate.". I suspect this may be an overstatement. To ascertain whether this claim is reasonable, the prevalence of mountain waves similar to those addressed in the present study would have to be taken into account. The tone of this remark could be moderated.

*We have moderated the tone in this statement according to the reviewer's suggestion. To our knowledge, this is the first study that estimates a radiative forcing in the context of a local mountain wave event. Since we have several mountain wave hot spots all over the world a study of the influence of mixing caused by these mountain waves on the radiation budget in the UTLS would be interesting.*

"The enhanced water vapor mixing ratios in the tropopause region strongly influences the radiative transfer in the UTLS. The estimated radiative forcing for our case, locally and temporally limited over the Southern Alps of New Zealand, exceeded 1 W m$^{-2}$ and suggests that mountain waves occurring in many locations all over the world may have a non-negligible effect on climate."

Page 28, Figure 4: I do not think the large negative flux of water vapour that can be seen in the bottom graph between x=-50 km and x=+50 km is discussed in sufficient detail in the text. This is an intriguing feature, which may seem puzzling to the reader. I advise the authors to include an interpretation of it, even if speculative, justifying its intensity, location and extent.

*We now have addressed this issue with an explanation regarding the tropospheric jet stream influence in this region. In Fig. 3 a decrease in the water vapor mixing ratio together with an increase in the potential temperature und changes in the horizontal wind components indicate an impact of the tropospheric jet stream between -50 and + 50 km distance that is also visible in the synoptic plots of Fig. 1. The vertical wind*

*component is not influenced by the jet stream. Since the decrease in water vapor cannot fully be eliminated by the used filter to obtain the water vapor perturbation we see this issue in the water vapor flux.*

"At the western edge of the mountains (between -50 and +30 km) we also observe a negative flux. This region is located in the vicinity of the tropospheric jet stream which influences the distribution of the water vapor mixing ratio by horizontal transport processes (Figure 3: decrease of $H_2O$ from west to east between -80 km and 0 km distance). This behavior cannot fully be eliminated by the used filter and is thus present in the water vapor perturbations by a few fluctuations with a negative weighting."

Page 30, Figure 6: In panel(a), the caption does not explain what the red dashed lines represent. Please add that information. In panel (b), the dotted line corresponding to the water vapour flux filtered for waves with wavelengths between 20 km and 80 km does not include a point at z=7.7 km, but the solid line does. Why is that? This choice should be justified convincingly in the text.

*We have added the definition for the red dashed lines. In panel (b) we now mark the two lines by different line character and different symbol to clarify that we show both wavelengths at each altitude point. At 7.7 km the values for both wavelengths are very similar which was not obvious in the first version of the plot.*

Technical corrections

Page 1, Line 13 in Abstract: "GV research aircraft". Does "GV" stand for anything, or is it just the name of the aircraft? In the first possibility, please expand and explain the acronym.

*The acronym is now explained in the text.*

Page 7, line 9: "q=H_2 O". If the two notations are equivalent, why use "H_2 O" instead of the shorter and more convenient notation "q", as is done throughout the manuscript? Is there any particular reason for this?

*We now use "q" in the whole method description section.*

Page 11, lines 29-30: "By the absence of vertical or horizontal transport and the existence of a well-mixed atmosphere, we are expecting no flux divergence.". This sentence does not sound very well. Consider rephrasing.

*We have rephrased the sentence.*

Page 25, Figure 1(b): Is this simply a magnification of Figure 1(a)? If yes, this should be mentioned in the caption.

*Fig. 1(b) is a magnification of Fig. 1(a) which is now mentioned in the caption.*

Page 26, caption of Figure 2: "... and topography are shown". It would be good to indicate what denotes the topography, i.e. the grey area at the bottom (as done in e.g. Figure 5).

*We have added this remark.*

Page 27, caption of Figure 3: "the diagonal blue dashed lines in the bottom panel display the phase shift between the vertical wind motion and perturbations in water vapor and theta". It is not totally clear what this means. Does the phase shift correspond to the horizontal distance between the bottom and top of these lines? Please clarify.

*We have rephrased the sentence to clarify the meaning of the phase shift.*